# A bacterial sulfonolipid triggers multicellular development in the closest living relatives of animals

Rosanna A Alegado[1†], Laura W Brown[2†], Shugeng Cao[2], Renee K Dermenjian[2], Richard Zuzow[3], Stephen R Fairclough[1], Jon Clardy[2*], Nicole King[1*]

[1]Department of Molecular and Cell Biology, University of California, Berkeley, Berkeley, United States; [2]Department of Biological Chemistry and Molecular Pharmacology, Harvard Medical School, Boston, United States; [3]Department of Biochemistry, Stanford University School of Medicine, Stanford, United States

**Abstract** Bacterially-produced small molecules exert profound influences on animal health, morphogenesis, and evolution through poorly understood mechanisms. In one of the closest living relatives of animals, the choanoflagellate *Salpingoeca rosetta*, we find that rosette colony development is induced by the prey bacterium *Algoriphagus machipongonensis* and its close relatives in the Bacteroidetes phylum. Here we show that a rosette inducing factor (RIF-1) produced by *A. machipongonensis* belongs to the small class of sulfonolipids, obscure relatives of the better known sphingolipids that play important roles in signal transmission in plants, animals, and fungi. RIF-1 has extraordinary potency (femtomolar, or $10^{-15}$ M) and *S. rosetta* can respond to it over a broad dynamic range—nine orders of magnitude. This study provides a prototypical example of bacterial sulfonolipids triggering eukaryotic morphogenesis and suggests molecular mechanisms through which bacteria may have contributed to the evolution of animals.

*For correspondence: jon_clardy@hms.harvard.edu (JC); nking@berkeley.edu (NK)

†These authors contributed equally to this work

## Introduction

Eukaryotes evolved in a world filled with bacteria and throughout their shared history these two branches of life have developed a complex set of ways to compete and cooperate with each other. While research on these interactions has historically emphasized bacterial pathogens, bacteria also regulate the biology of eukaryotes in many other ways (*McFall-Ngai 1999*; *Koropatnick et al. 2004*; *Mazmanian et al. 2005*; *Falkow 2006*; *Hughes and Sperandio 2008*; *Desbrosses and Stougaard 2011*) and may have exerted critical influences on animal evolution. Choanoflagellates, microscopic bacteria-eating eukaryotes that are the closest living relatives of animals (*James-Clark 1868*; *Saville Kent 1880*; *Hibberd 1975*; *Carr et al. 2008*; *King et al. 2008*; *Ruiz-Trillo et al. 2008*), could provide particularly important insights into the mechanisms underlying bacterial influences on animal biology and evolution. Moreover, some choanoflagellates have both solitary and multicellular stages in their life histories (*Leadbeater 1983*; *Karpov and Coupe 1998*; *Dayel et al. 2011*) and understanding the environmental cues that regulate choanoflagellate colony formation could provide a molecular model for animal multicellularity.

## Results

In the choanoflagellate *Salpingoeca rosetta*, rosette-shaped multicellular colonies develop when a single founder cell undergoes multiple rounds of incomplete cytokinesis, leaving neighboring cells physically attached by fine intercellular bridges (*Fairclough et al. 2010*; *Dayel et al. 2011*). Although the original stock of *S. rosetta* (ATCC50818) was established from a rosette colony (*Dayel et al.*

**eLife digest** All animals, including humans, evolved in a world filled with bacteria. Although bacteria are most familiar as pathogens, some bacteria produce small molecules that are essential for the biology of animals and other eukaryotes, although the details of the ways in which these bacterial molecules are beneficial are not well understood.

The choanoflagellates are water-dwelling organisms that use their whip-like flagella to move around, feeding on bacteria. They can exist as one cell or a colony of multiple cells and, perhaps surprisingly, are the closest known living relatives of animals. This means that experiments on these organisms have the potential to improve our understanding of animal development and the transition from egg to embryo to adult.

Alegado *et al.* have explored how the morphology of *Salpingoeca rosetta,* a colony-forming choanoflagellate, is influenced by its interactions with various species of bacteria. In particular, they find that the development of multicellularity in *S. rosetta* is triggered by the presence of the bacterium *Algoriphagus machipongonensis* as well as its close relatives. They also identify the signaling molecule produced by the bacteria to be $C_{32}H_{64}NO_7S$; this lipid molecule is an obscure relative of the sphingolipid molecules that have important roles in signal transmission in animals, plants, and fungi. Moreover, Alegado *et al.* show that *S. rosetta* can respond to this molecule – which they call rosette-inducing factor (RIF-1) – over a wide range of concentrations, including concentrations as low as $10^{-17}$ M.

The work of Alegado *et al.* suggests that interactions between *S. rosetta* and *Algoriphagus* bacteria could be a productive model system for studying the influences of bacteria on animal cell biology, and for investigating the mechanisms of signal delivery and reception. Moreover, the molecular mechanisms revealed by this work leave open the possibility that bacteria might have contributed to the evolution of multicellularity in animals.

*2011*), laboratory cultures consistently produced single cells, with small numbers of rosette colonies forming only sporadically (*Figure 1A*, *Figure 1—figure supplement 1*). Serendipitously, we discovered that the bacterial community influences rosette colony development. Treatment of the ATCC50818 culture with an antibiotic cocktail resulted in a culture of *S. rosetta* cells that proliferated robustly by feeding on the remaining antibiotic-resistant bacteria but never formed rosette colonies, even upon removal of antibiotics (*Figure 1B*). This culture line is hereafter referred to as RCA (for 'Rosette Colonies Absent'). Supplementation of RCA cultures with bacteria from ATCC50818 restored rosette colony development, revealing that *S. rosetta* cells in the RCA culture remained competent to form colonies and would do so when stimulated by the original community of environmental bacteria.

To determine which co-isolated bacterial species stimulate rosette colony development in *S. rosetta*, the RCA cell line was supplemented with 64 independent bacterial isolates from ATCC50818 and monitored for the appearance of rosette colonies. Only one bacterial species from ATCC50818, the previously undescribed *Algoriphagus machipongonensis* (phylum Bacteroidetes; *Alegado et al. 2012*), induced rosette colony development in the RCA cell line (*Figure 1C*). *S. rosetta* cultures fed solely with *A. machipongonensis* yielded high percentages of rosette colonies (*Figure 1—figure supplement 1*), demonstrating that no other co-isolated bacterial species is required to stimulate rosette colony development.

What was not clear was whether other bacteria might also be competent to induce rosette colony development. Therefore, representative Bacteroidetes and non-Bacteroidetes bacteria were grown and fed to RCA cultures (*Figure 2*, *Table 1*). None of the non-Bacteroidetes species tested, including members of the γ-proteobacteria, α-proteobacteria, and Gram-positive bacteria, were competent to induce rosette colony development. In contrast, all 15 *Algoriphagus* species tested induced rosette colony development, as did six of 16 other closely related species tested in the Bacteroidetes phylum (*Table 1*). Therefore, the ability to induce rosette colony development is enriched in *Algoriphagus* bacteria and their relatives.

Although Bacteroidetes bacteria regulate morphogenetic processes in such diverse lineages as animals, red algae, and green algae (*Provasoli and Pintner 1980*; *Matsuo et al. 2005*; *Mazmanian*

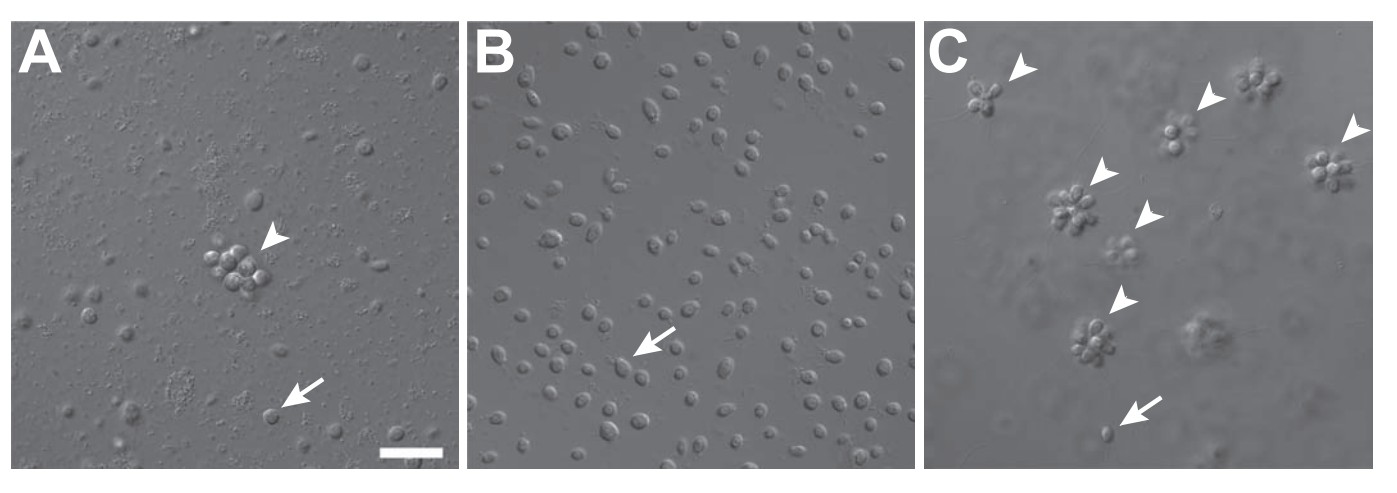

**Figure 1**. Rosette colony development in *S. rosetta* is regulated by *A. machipongonensis*. (**A**) The original culture of *S. rosetta*, ATCC 50818, contains diverse co-isolated environmental bacteria and forms rosette colonies (arrowheads) rarely. (**B**) Treatment of ATCC50818 with a cocktail of antibiotics reduced the bacterial diversity and yielded an *S. rosetta* culture line, RCA, in which rosette colonies never formed. (Representative single cells indicated by arrows.) (**C**) Addition of *A. machipongonensis* to RCA cultures was sufficient to induce rosette development. Scale bar, 2 μm.

The following figure supplements are available for figure 1:

**Figure supplement 1**. Frequency of rosette colonies in *S. rosetta* environmental isolate ATCC 50818, RCA with and without *A. machipongonensis* and a monoxenic line with *A. machipongonensis* feeder bacteria (Px1).

*et al. 2005*), the bacterially produced chemical cues that regulate most of these partnerships remain obscure. The limited phylogenetic distribution of bacteria capable of inducing rosette colony development suggested that *A. machipongonensis* and its close relatives may produce a characteristic molecule that could be identified biochemically. The complete absence of rosette colonies in RCA cultures provided the basis for a robust bioassay that we developed to identify the rosette-inducing molecule(s), which we named RIFs (Rosette-Inducing Factors), from *A. machipongonensis* cultures. Preliminary studies demonstrated that RIF activity was present in conditioned medium from *A. machipongonensis*, even when grown in the absence of choanoflagellates. Furthermore, the activity was also found in the *A. machipongonensis* cell envelope and was heat, protease, and nuclease resistant, revealing that the compound is not a protein, RNA, or DNA (*Table 2*).

The bacterial cell envelope components lipopolysaccharide (LPS) and peptidoglycan (PGN) from Gram-negative bacteria have long been known to affect host biology (*Cohn and Morse 1960*; *Hoffmann et al. 1999*; *Kopp and Medzhitov 1999*; *Takeuchi et al. 1999*; *Medzhitov and Janeway 2000*; *Kimbrell and Beutler 2001*; *Koropatnick et al. 2004*), but neither LPS nor PGN from *A. machipongonensis* triggered rosette development, alone or in combination (*Figure 3A*). Instead, we found that *A. machipongonensis* crude lipid extracts enriched in sphingolipids robustly induced rosette development (*Figure 3A*). In animals, sphingolipid signaling pathways regulate developmental processes such as cell death, survival, differentiation, and migration (*Prieschl and Baumruker 2000*; *Pyne and Pyne 2000*; *Spiegel and Milstien 2000*; *Hannun et al. 2001*; *Merrill et al. 2001*; *Herr et al. 2003*). Moreover, sphingolipids serve essential functions both as structural components of cell membranes and as signaling molecules in diverse eukaryotes (*Hannich et al. 2011*). In contrast, the phylogenetic distribution of sphingolipids in bacteria is largely limited to Bacteroidetes and *Sphingomonas*, where their endogenous functions are poorly understood (*Olsen and Jantzen 2001*; *An et al. 2011*).

To isolate and characterize the molecule(s) underlying RIF activity, we focused on the fraction enriched in sphingolipids. Lipids isolated from 160 L of *A. machipongonensis* culture were separated using preparative liquid chromatography–mass spectrometry (LC-MS) and the activity of each fraction was measured using the rosette colony induction bioassay. RIF activity tracked with a single fraction, which was further purified by several rounds of preparative thin-layer chromatography (*Figure 3— figure supplement 1*) to yield approximately 700 μg of active compound (RIF-1) with sufficient purity

**Table 1.** Species tested for colony induction

| Species | 16S rDNA accession number | Reference | Rosette colonies |
|---|---|---|---|
| *Algoriphagus machipongonensis* PR1 | NZ_AAXU00000000 | *Alegado et al. (2012)* | + |
| *Algoriphagus alkaliphilus* AC-74 | AJ717393 | *Tiago et al. (2006)* | + |
| *Algoriphagus boritolerans* T-22 | AB197852 | *Ahmed et al. (2007)* | + |
| *Algoriphagus mannitolivorans* JC2050 | AY264838 | *Yi and Chun (2004)* | + |
| *Algoriphagus marincola* SW-2 | AY533663 | *Yoon et al. (2004)* | + |
| *Algoriphagus ornithinivorans* JC2052 | AY264840 | *Yi and Chun (2004)* | + |
| *Algoriphagus vanfongensis* KMM 6241 | EF392675 | *Van Trappen et al. (2004)* | + |
| *Algoriphagus antarcticus* LMG 21980 | AJ577141 | *Nedashkovskaya et al. (2004)* | + |
| *Algoriphagus aquimarinus* LMG 21971 | AJ575264 | *Nedashkovskaya et al. (2004)* | + |
| *Algoriphagus chordae* LMG 21970 | AJ575265 | *Nedashkovskaya et al. (2004)* | + |
| *Algoriphagus halophilus* JC2051 | AY264839 | *Yi and Chun (2004)* | + |
| *Algoriphagus locisalis* MSS-170 | AY835922 | *Yoon et al. (2005a)* | + |
| *Algoriphagus ratkowskyi* LMG 21435 | AJ608641 | *Bowman et al. (2003)* | + |
| *Algoriphagus terrigena* DS-44 | DQ178979 | *Yoon et al. (2006)* | + |
| *Algoriphagus winogradskyi* LMG 21969 | AJ575263 | *Nedashkovskaya et al. (2004)* | + |
| *Algoriphagus yeomjeoni* MSS-160 | AY699794 | *Yoon et al. (2005b)* | + |
| *Agrobacterium tumefaciens* C58 | AE007870 | *Wood et al. (2001)* | + |
| *Aquiflexum balticum* BA160 | AJ744861 | *Brettar et al. (2004a)* | − |
| *Bacillus subtilis* 168 | AL009126 | *Kunst et al. (1997); Burkholder and Giles (1947); Spizizen (1958)* | − |
| *Bacteroides fragilis* NCTC9343 | CR626927 | *Cerdeno-Tarraga et al. (2005)* | − |
| *Belliella baltica* BA134 | AJ564643 | *Brettar et al. (2004b)* | − |
| *Caulobacter crescentus* CB15 | AE005673 | *Nierman et al. (2001)* | − |
| *Croceibacter atlanticus* HTCC2559 | NR_029064 | *Cho and Giovannoni (2003)* | − |
| *Cyclobacterium marinum* LMG 13164 | AJ575266 | *Raj and Maloy 1990)* | + |
| *Cytophaga hutchinsonii* ATCC 33406 | M58768 | *Lewin (1969)* | + |
| *Dyadobacter fermentans* DSM 18053 | NR_027533 | *Chelius and Triplett (2000)* | + |
| *Echinicola pacifica* KMM 6172 | NR_043619 | *Nedashkovskaya et al. (2006)* | − |
| *Escherichia coli* MG1655 | U00096 | *Blattner et al. (1997)* | − |
| *Flavobacteria johnsoniae* UW101 | CP000685 | *Bernardet et al. (1996)* | − |
| *Flectobacillus major* DSM 103 | M62787 | *Raj and Maloy (1990)* | + |
| *Listeria monocytogenes* 10403S | CP002002 | *Edman et al. (1968)* | − |
| *Magnetospirillum magneticum* AMB-1 | AP007255 | *Matsunaga et al. (2005)* | − |
| *Microscilla marina* ATCC 23134 | M123134 | *Garrity (2010)* | − |
| *Oceanostipes pacificus* HTCC2170 | CP002157 | *Oh et al. (2011)* | − |
| *Robiginitalea biformata* HTCC2501 | CP001712 | *Cho and Giovannoni (2004)* | − |
| *Salinibacter ruber* DSM13855 | CP000159 | *Anton et al. (2002)* | − |
| *Sphingomonas wittichii* RW1 | CP000699 | *Miller et al. (2010)* | − |
| *Vibrio fischeri* ES114 | CP000021 | *Ruby et al. (2005)* | − |
| *Zobellia galactonovorans* Dsij | NR_025053 | *Barbeyron et al. (2001)* | + |
| *Zobellia uliginosa* ATCC 14397 | M62799 | *Matsuo et al. (2003)* | + |

−: no rosette colonies observed; +: rosette colonies observed

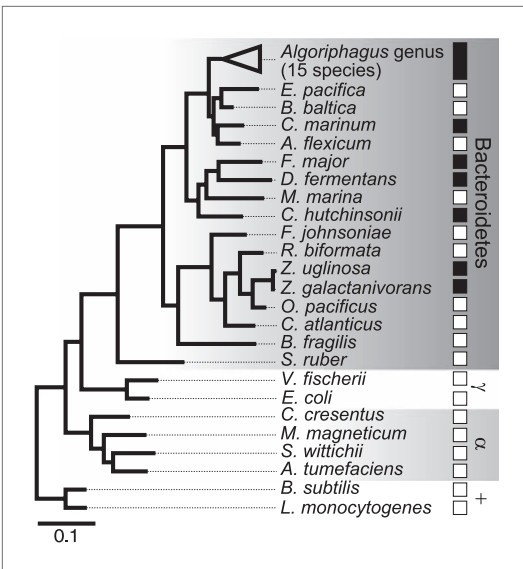

**Figure 2**. Diverse members of the Bacteroidetes phylum induce rosette colony development. A maximum likelihood phylogeny inferred from 16S rDNA gene sequences reveals the evolutionary relationships among *A. machipongonensis*, other members of the Bacteroidetes phylum, and representative γ-proteobacteria (γ), α-proteobacteria (α), and Gram-positive (+) bacteria. All 15 members of the *Algoriphagus* genus (**Table 1**), as well as six other species in the Bacteroidetes phylum, were competent to induce colony development (filled squares). In contrast, no species outside of Bacteroidetes and most of the non-*Algoriphagus* bacteria tested failed to induce rosette colony development (open squares). Scale bar, 0.1 substitutions per nucleotide position. DOI: 10.7554/eLife.00013.006

for structural analysis. RIF-1 represents only 0.015% of the *A. machipogonensis* sphingolipid pool. Based on high-resolution mass spectrometry, RIF-1 has a molecular formula of $C_{32}H_{64}NO_7S$ (M-H: exptl. 606.44027, calcd. 606.44035, **Figure 3—figure supplement 2**). Detailed analysis of one- and two-dimensional (COSY, HMBC, TOCSY, **Figure 3—figure supplements 3–15**) nuclear magnetic resonance (NMR, 600 MHz, **Table 3**) spectra revealed the planar structure of RIF-1, an unusual sulfonolipid shown in **Figure 3B**.

Sulfonolipids like RIF-1 resemble sphingolipids, but there are important differences between the two. In sphingolipids, a sphingoid base (1,3-dihydroxy-2-aminoalkane) is linked through an amide bond to a fatty acid. The long alkyl chains of both the sphingoid base and fatty acid vary in length, branching, number of double bonds, and placement of hydroxyl substituents. In RIF-1, a capnoid base (2-amino-3-hydroxy-15-methyl-1-*sulfonic acid*) replaces the sphingoid base, and the sphingolipid hydroxyl, which is the attachment point for the major diversifying elements of the sphingolipid family, is replaced by a sulfonic acid. While members of the sphingolipid family, such as the ceramides, glycosphingolipids, sphingomyelins, and gangliosides, differ by the groups attached to the hydroxyl, the sulfonic acid function in sulfonolipids like RIF-1 has no reported diversifying modifications. In this sense sulfonolipid diversity appears more limited than sphingolipid diversity. Significantly, commercial sphingolipids (sphingomyelin, monosiloganglioside, galactocerebroside, and N-palmitoyl-DL-dihydrolacto cerebroside; **Table 2**) failed to show any activity in our assay system. To our knowledge, RIF-1 is the first sulfonolipid demonstrated to influence developmental processes in eukaryotes.

Finally, we investigated the potency of RIF-1 and its ability to induce colony development under plausible environmental conditions. Purified RIF-1 induces rosette formation with a bell-shaped dose-response curve over a broad range of concentrations, from $10^{-2}$ to $10^7$ fM or some nine orders of magnitude (**Figure 4**). No observable effects were seen below $10^{-5}$ fM, and RIF-1 appears to be inactive above $10^8$ fM. *A. machipongonensis* conditioned medium contains $10^4$ fM RIF-1 (**Figure 4—figure supplements 1–3**), and even if this conditioned medium measurement exaggerates natural concentrations by a factor of $10^6$, *S. rosetta* could still respond to its presence. The shape of the dose-response curve and the potency of RIF-1 suggest that *S. rosetta* perceives RIF-1 in a manner consistent with a receptor-ligand interaction, albeit a receptor of exquisite sensitivity and remarkable dynamic range. While RIF-1 is the only molecule detected with rosette-inducing activity, its maximal activity (5.6 ± 0.5% colonial cells/total cells) differs from that of the sphingolipid-enriched lipid fraction (19.2 ± 4.6% colonial cells/total cells; **Figure 4**). This difference may be due to delivery issues of the purified and highly hydrophobic molecule, which in nature resides in membranes and potentially in membrane vesicles. Alternatively, the full potency of RIF-1 as an inducer of colony development may require additional *A. machipongonensis* molecules not identified in this study.

## Discussion

These data reveal that RIF-1, a sulfonolipid produced by the prey bacterium *A. machipongonensis*, regulates morphogenesis in its predator, *S. rosetta*. The ecological relevance of this signaling

**Table 2.** Responses of RCA culture to various supplements

| Treatment | Rosette colonies | Interpretation |
|---|---|---|
| Sea water | − | |
| CM from ATCC50818 | + | RIF-1 present in environmental isolate ATCC 50818 |
| CM from RCA | − | RIF-1 is absent in RCA lines |
| Live *A. machipongonensis* (cell pellet) | ++ | RIF-1 is produced by *A. machipongonensis* |
| Heat killed *A. machipongonensis* (cell pellet) | ++ | RIF-1 is resistant to heat |
| *A. machipongonensis* CM | + | RIF-1 is released by live *Algoriphagus* |
| *A. machipongonensis* CM, boiled 10 min | + | RIF-1 is not labile |
| *A. machipongonensis* CM + Proteinase K | + | RIF-1 is not a protein |
| *A. machipongonensis* CM + DNAse | + | RIF-1 is not DNA |
| *A. machipongonensis* CM + RNAse | + | RIF-1 is not RNA |
| *A. machipongonensis* CM, MeOH extract | + | RIF-1 is an organic compound |
| *A. machipongonensis* cell pellet, MeOH extract | ++ | RIF-1 is present in the *Algoriphagus* cell envelope |
| *A. machipongonensis* cell pellet, Bligh-Dyer extract | ++ | RIF-1 is a lipid |
| Sphingomyelin (20 mg mL$^{-1}$) | − | Sphingomyelin does not induce rosette colony development |
| Monosialoganglioside (20 mg mL$^{-1}$) | − | Monosialoganglioside does not induce rosette colony development |
| Galactocerebroside (20 mg mL$^{-1}$) | − | Galactocerebroside does not induce rosette colony development |
| N-palmitoyl-DL-dihydrolacto cerebroside (20 mg mL$^{-1}$) | − | N-palmitoyl-DL-dihydrolacto cerebroside does not induce rosette colony development |

−: no induction; +: low induction; ++: high induction

CM: conditioned medium; RCA: rosette colonies absent; RIF-1: rosette inducing factor 1

interaction is indicated both by the coexistence of *S. rosetta* and *A. machipongonensis* in nature and by the fact that the activity of RIF-1 at femtomolar concentrations makes it markedly more potent than other marine signaling molecules [e.g., *Vibrio* autoinducer (**Schaefer et al. 1996**) and the tripeptide pheromones of the Caribbean spiny lobster (**Ziegler and Forward 2007**)]. The potency of RIF-1 signaling compares favorably with that of silkworm moth sex pheromone signaling, in which vapors from an ~4 fM solution of bombykol, the sex pheromone of the silkworm moth, induce a pronounced wing fluttering response in males (**Butenandt et al. 1961**; **Agosta 1992**; **Roelofs 1995**). While it is formally possible that RIF-1-dependent rosette colony development is a promiscuous response to sphingolipid-type molecules, only a handful of sulfonolipids like RIF-1 have been reported (**Godchaux and Leadbetter 1980**, **1983**, **1984**, **1988**; **Drijber and McGill 1994**; **Kamiyama et al. 1995a**, **1995b**; **Kobayashi et al. 1995**) and no other *A. machipongonensis* lipid tested in this study induced rosette colony development. Therefore we favor a model in which *A. machipongonensis* cell density, as revealed by RIF-1 concentration, provides *S. rosetta* with an indication of conditions under which rosette colony development would be advantageous, for instance by promoting more efficient capture of planktonic bacteria (**Kreft 2010**). In analogy to the chemotaxis system of bacteria (**Falke et al. 1997**), the ability of *S. rosetta* to respond to increasing bacterial cell density likely requires the hypothesized RIF-1 receptor to become less sensitive at higher concentrations. The high concentration cutoff in the dose-response curve reflects a complete loss of sensitivity at high, but non-physiological, RIF-1 concentrations. Although the presence of RIF-1 in the *A. machipongonensis* cell envelope suggests that it may be encountered by *S. rosetta* during phagocytosis, it can also function at a distance. We hypothesize that RIF-1 may be released into the environment in membrane vesicles, which have been described in Gram-negative bacteria such as Bacteroidetes (**Zhou et al. 1998**; **Møller et al. 2005**), and that additional membrane constituents might be required for the full potency of RIF-1. In the future,

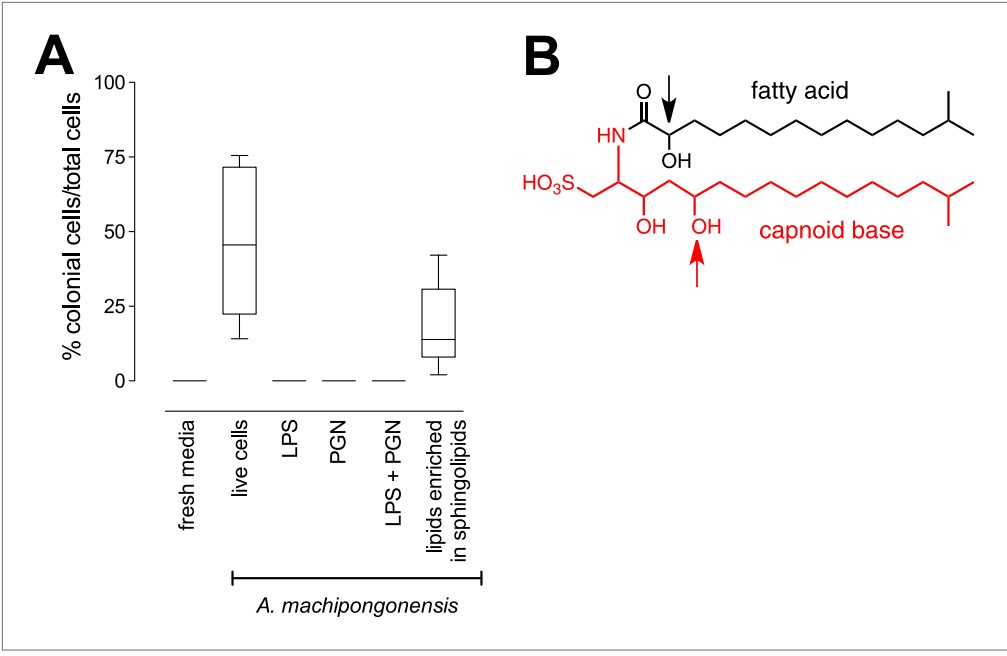

**Figure 3**. RIF-1, a sulfonolipid that induces rosette colony development. (**A**) Rosette colony development is induced by live *A. machipongonensis* and the sphingolipid-enriched lipid fraction (20 mg mL$^{-1}$), but not by fresh medium, *A. machipongonensis* LPS (10 mg mL$^{-1}$), PGN (50 mg mL$^{-1}$), or LPS+PGN. Shown are the whisker-box plots of the % colonial cells/total cells under each condition in three independent experiments. (**B**) The molecular structure of RIF-1 deduced from MS and 1D- and 2D-NMR data. The RIF-1 structure, 3,5-dihydroxy-2-(2-hydroxy-13-methyltetradecanamido)-15-methylhexadecane-1-sulfonic acid, has two parts: a base (shown in red) that defines the capnine, and a fatty acid (shown in black). Features that distinguish RIF-1 from other known capnoids are shown with colored arrows: the 2-hydroxy on the fatty acid (black) and the 5-hydroxy on the capnine base (red).

The following figure supplements are available for figure 3:

**Figure supplement 1**. Separation of *A. machipongonensis* sphingolipids by thin layer chromatography (TLC).

**Figure supplement 2**. MS/MS analysis of RIF-1.

**Figure supplement 3**. Key two-dimensional (2D) correlations of RIF-1: Observed COSY correlations.

**Figure supplement 4**. Key two-dimensional (2D) correlations of RIF-1: Observed HMBC spin system.

**Figure supplement 5**. Key two-dimensional (2D) correlations of RIF-1: Observed TOCSY spin system.

**Figure supplement 6**. $^1$H NMR spectrum of RIF-1.

**Figure supplement 7**. gHMQC spectrum of RIF-1.

**Figure supplement 8**. gCOSY spectrum of RIF-1.

**Figure supplement 9**. Expanded dqfCOSY spectrum of RIF-1.

**Figure supplement 10**. Expanded dqfCOSY spectra of RIF-1.

**Figure supplement 11**. gHMBC spectrum of RIF-1.

**Figure supplement 12**. Expanded gHMBC spectrum of RIF-1 ($\delta_H$ 0–4.00 ppm/$\delta_C$ 15.0–85.0 ppm).

*Figure 3. Continued on next page*

*Figure 3. Continued*

**Figure supplement 13**. Expanded gHMBC spectrum of RIF-1 ($\delta_H$ 0.80–1.80 ppm/$\delta_C$ 20.0–40.0 ppm).

**Figure supplement 14**. TOCSY spectrum of RIF-1.

**Figure supplement 15**. Expanded TOCSY spectrum of RIF-1 ($\delta_H$ 0.50–4.25 ppm/$\delta_C$ 0.50–4.25 ppm).

elucidating RIF-1 delivery, along with determining the three-dimensional structure of RIF-1 and characterizing sulfonolipids from other Bacteroidetes will begin to provide the needed foundation for a molecular understanding of how *S. rosetta* perceives RIFs.

The morphogenetic interaction described here between *S. rosetta* and *A. machipongonensis* raises the possibility that bacterially-produced sphingolipids in general, and sulfonolipids in particular, may be essential for the chemical signaling that allows Bacteroidetes to influence cell differentiation and morphogenesis in diverse animals (*Falkow 2006*; *Mazmanian et al. 2008*; *Lee and Mazmanian 2010*; *An et al. 2011*). Sulfonolipids like RIF-1 have been reported to have therapeutic activities, but their endogenous functions are not known. Sulfobacins A and B, which were isolated from the culture broth of a *Chryseobacterium* sp. were reported as von Willebrand factor receptor antagonists, and flavocristamide A, from a related bacterial species, was reported as a DNA polymerase α inhibitor (*Kamiyama et al. 1995a*; *Kobayashi et al. 1995*). The pervasiveness of interactions between Bacteroidetes and animals (*Webster et al. 2004*; *Wexler 2007*), coupled with the close evolutionary relationship between choanoflagellates and animals (*King and Carroll 2001*; *King 2004*; *King et al. 2008*; *Ruiz-Trillo et al. 2008*), raise the possibility that the connection between Bacteroidetes and animal development has deep evolutionary roots (*McFall-Ngai 1999*). The discovery of RIF-1 and its biological activity toward *S. rosetta* provides both the molecular basis and model organism for further understanding a new and potentially important class of small molecule information transfer.

## Materials and methods

### Choanoflagellate husbandry and microscopy

The environmental isolate of *Salpingoeca rosetta* is deposited at the American Tissue Culture Collection (ATCC) under the designation ATCC50818 (*King et al. 2003*). The Rosette Colonies Absent (RCA) culture line was produced from ATCC50818 by serial treatment with chloramphenicol (68 µg mL$^{-1}$), ampicillin (50 µg mL$^{-1}$), streptomycin (50 µg mL$^{-1}$), and erythromycin (50 µg mL$^{-1}$) (*Fairclough et al. 2010*). A monoxenic line of *S. rosetta* (Px1) was generated by treating ATCC 50818 with a combination of ofloxacin (10 µg mL$^{-1}$), kanamycin (50 µg mL$^{-1}$), and streptomycin (50 µg mL$^{-1}$) antibiotics to kill the undefined environmental bacteria. Following several rounds of serial dilution, a single cell was isolated by FACS and supplemented with *A. machipongonensis* (*Dayel et al. 2011*). All three *S. rosetta* cell lines (ATCC 50818, RCA, and Px1) were grown in cereal grass infused seawater at 25°C and maintained by splitting cultures 1:10 into fresh medium every 3 days (*King et al. 2009*). Live cells were imaged with a Leica DMI6000B microscope equipped with a DFC350 FX camera.

### Bioassay for rosette colony development

Under laboratory conditions, *S. rosetta* differentiates into a variety of cell types including attached thecate cells, solitary swimmers, rosette colonies, chain colonies and loose, disorganized associations of cells attached to one another at the collar or to bacterial biofilms (*Dayel et al. 2011*). *S. rosetta* rosette colonies can be distinguished from other cell types in that they contain clusters of at least four closely associated cells with organized polarity; each cell oriented with its flagellum pointing outward from a central focus. In the qualitative bioassay, RCA cultures were diluted in fresh medium to a density of approximately $10^4$–$10^5$ cells mL$^{-1}$, aliquoted into 24-well flat bottom culture dishes (Costar, Corning, NY, USA), supplemented with various treatments, and scored for the presence or absence of rosette colonies after 48 hr. For quantitative measurements, RCA cultures were diluted as before into six-well flat bottom culture dishes. To measure the percentage of cells within rosette colonies, each well was scraped to detach thecate cells and the total number of cells and the total number of cells in each rosette colony were counted with a Bright-Line hemacytometer (Hausser Scientific, Horsham, PA, USA).

**Table 3.** Table of NMR chemical shifts

| Position | δ ¹H (multiplicity, J, #H) | ¹³C (δ, ppm) |
|---|---|---|
| NH | 8.21 (d, J=9.1 Hz, 1H) | |
| 1 | 3.01 (dd, J=14.2, 5.1 Hz, 1H) | 51.87 |
| | 2.56 (dd, J=14.3, 3.6 Hz, 1H) | |
| 2 | 3.88 (ddd, J=13.0, 8.6, 4.5 Hz, 1H) | 50.89 |
| 3 | 3.78 – 3.71 (m, 1H) | 71.51 |
| OH3 | 5.20 (d, J=4.2 Hz, 1H) | |
| 4 | 1.51 – 1.47 (m, 1H) | 41.36 |
| | 1.33 – 1.29 (m, 1H) | |
| 5 | 3.61 – 3.53 (m, 1H) | 70.20 |
| OH5 | 4.31 (d, J=3.5 Hz, 1H) | |
| 6 | 1.34 – 1.29 (m, 1H) | 37.27 |
| | 1.24 – 1.20 (m, 1H) | |
| 7–13 | 1.21 – 1.27 (br s, 14H) | 22.5–29.6 |
| 14 | 1.16 – 1.11 (m, 2H) | 38.91 |
| 15 | 1.52 – 1.47 (m, 1H) | 27.79 |
| 16, 17 | 0.84 (d, J=6.6 Hz, 6H) | 22.09 |
| 1′ | | 173.23 |
| 2′ | 3.80 (dd, J=6.6, 4.2 Hz, 1H) | 71.29 |
| OH2′ | 5.52 (d, J=5.0 Hz, 1H) | |
| 3′ | 1.59 – 1.54 (m, 1H) | 34.85 |
| | 1.50 – 1.45 (m, 1H) | |
| 4′? | 1.35 – 1.30 (m, 2H) | 24.99 |
| 5′–11′ | 1.21 – 1.27 (brs, 14H) | 22.5–29.6 |
| 12′ | 1.16 – 1.11 (m, 2H) | 38.91 |
| 13′ | 1.52 – 1.47 (m, 1H) | 27.79 |
| 14′, 15′ | 0.84 (d, J=6.6 Hz, 6H) | 22.09 |

## Isolation and identification of *A. machipongonensis*

A partial representation of the bacterial flora from ATCC50818 was isolated by standard dilution-plating technique on modified Zobell medium agar (*Carlucci and Pramer 1957*) at 25°C. Individual isolates were tested for their morphogenic activity by supplementing RCA cultures with a single colony of each isolate. Of 64 isolates tested, the only one that restored rosette colony development to the RCA cell line was a species that formed pink-pigmented colonies (designated strain PR1). Strain PR1 was used to inoculate liquid modified Zobell medium at 25°C and grown with aeration overnight. PR1 cells were harvested by centrifugation, and genomic DNA was isolated using a Bacterial Genomic DNA Mini-prep Kit (Bay Gene, Burlingame, CA, USA) according to the manufacturer's specifications. The 16S rRNA gene was amplified using universal primers 8F (5′-AGAGTTTGATCCTGGCTCAG-3′) and 1492R (5′-ACCTTGTTACGRCTT-3′) (*Weisburg et al. 1991*); comparison of the PR1 16S rRNA sequence to the Greengenes 16S rRNA database (*DeSantis et al. 2006*) revealed strain PR1 to be most closely related to members of the *Algoriphagus* genus within the Bacteroidetes phylum. PR1 was subsequently named *Algoriphagus machipongonensis* (*Bradley et al. 2009*).

## Generating a phylogenetic framework for testing the diversity of bacteria that induce rosette colony development

To investigate whether the ability to trigger rosette colony development was specific to *A. machipongonensis*, we tested three classes of bacterial species for their morphogenic capacity: 15 species in the *Algoriphagus* genus, 16 non-*Algoriphagus* members of the Bacteroidetes phylum, and eight species representing three additional major clades within Bacteria. Each species was screened for morphogenic activity using the bioassay for rosette colony development. Live cells from individual colonies grown from solid agar plates were added directly to RCA cultures and scored for the presence or absence of rosette colonies 48 hr after inoculation. Each bacterial species was tested three times.

To determine the phylogenetic distribution of morphogenic activity in the bacterial species tested (*Table 1*), a sequence alignment of 16S rDNA genes from each species was generated by iterative pairwise comparisons using FSA (*Bradley et al. 2009*). Poorly aligned regions were removed by Gblocks version 0.91b (*Castresana 2000*; *Talavera and Castresana 2007*) using default block parameters. A distance matrix (distance options according to the Kimura two-parameter model), including clustering with the maximum likelihood algorithm, was calculated using Phylip version 3.67 (*Falenstein 1989*). Support for the resulting tree topology was estimated using bootstrap analysis (1000 replicates).

## Biochemical analysis of Rosette Inducing Factor (RIF-1)

To determine the biochemical nature of RIF-1, *A. machipongonensis* cell fractions and conditioned medium were subjected to a battery of treatments. The results of these tests are summarized in *Table 2*. Conditioned medium (CM) was generated by pelleting either choanoflagellates grown

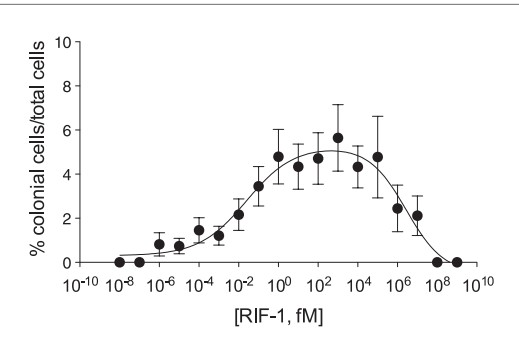

**Figure 4**. Purified RIF-1 is active at plausible environmental concentrations. RIF-1 concentrations ranging from $10^{-2}$ to $10^{7}$ fM induce rosette colony development in RCA cultures. Frequency of rosette colony development was quantified in RCA cultures 2 days after treatment with a dilution series of purified RIF-1. Data are mean ± s.e. from three independent experiments. Line indicates non-linear regression of the RIF-1 activity profile.

The following figure supplements are available for figure 4:

**Figure supplement 1**. Detection of purified RIF-1.

**Figure supplement 2**. Detection of RIF-1 in the conditioned medium of *A. machipongonensis*.

**Figure supplement 3**. Co-injection of concentrated conditioned medium with purified RIF-1.

in cereal grass infused with seawater or *A. machipongonensis* cultures grown in seawater complete medium (*Atlas 2004*) and filtering the culture supernatant through a 0.22 µm pore filter (Millipore) to remove live bacteria. To test whether RIF-1 activity required live bacteria, *A. machipongonensis* was grown overnight at 25°C, centrifuged at 16,000×*g* for 1 min to pellet cells, and heated for 30 min at 80°C to kill viable bacteria. To test whether RIF-1 activity might be heat labile (e.g., a polypeptide), *A. machipongonensis* CM was boiled for 10 min. To test whether RIF-1 was a protein, *A. machipongonensis* CM was incubated with 200 µg mL$^{-1}$ proteinase K (New England Biolabs, Ipswich, MA, USA) for 2 hr at 37°C. To test whether RIF-1 was a nucleic acid, 25 mL of *A. machipongonensis* CM was lyophilized, resuspended in 2.5 mL of water, and extracted with 100% ethanol to a final concentration of 80% (vol/vol) for 2 hr at −20°C and the precipitate was collected by centrifugation for 30 min at 4000×*g* at 4°C. The precipitate was dissolved in 0.01 M PBS (containing 10 mM MgCl$_2$ and 1 mM CaCl$_2$) and incubated with either RNase A (100 µg mL$^{-1}$; Sigma) or DNase I (100 µg mL$^{-1}$; Sigma) for 2.5 hr at 37°C. To test whether RIF-1 activity was in the methanolic extract, *A. machipongonensis* cell pellet and CM were lyophilized and vortexed with methanol. Each suspension was centrifuged at 8000 rpm for 5 min and the methanol layer recovered and dried.

To test whether RIF-1 was a lipid, *A. machipongonensis* cell pellet was extracted according to the Bligh–Dyer method (*Bligh and Dyer 1959*). Briefly, the cell pellet was resuspended in 3 vol of 1:2 (vol/vol) CHCl$_3$:MeOH and vortexed. One volume of CHCl$_3$ was added, and the mixture vortexed. One volume of distilled water was then added, and the mixture vortexed. The same was then centrifuged at 1000 rpm for 5 min and the bottom layer recovered and dried.

To test whether RIF-1 was a component of lipopolysaccharide (LPS), *A. machipongonensis* LPS was isolated using a method from *Apicella (2008)*. Lyophilized *A. machipongonensis* cells were ground with a mortar and pestle and suspended in 10 mM Tris–Cl buffer (pH 8.0), containing 2% sodium dodecyl sulfate (SDS), 4% 2-mercaptoethanol, and 2 mM MgCl$_2$. The mixture was vortexed and incubated at 65°C until solubilized. Proteinase K (20 mg mL$^{-1}$) was added to the mixture, and incubated at 65°C for an additional hour, followed by 37°C incubation overnight. 3 M sodium acetate was then added and the sample mixed. Following addition of cold absolute ethanol to the cell suspension, the sample was incubated overnight at –20°C to allow precipitate to form. The mixture was centrifuged at 4000×*g* for 15 min, and the supernatant discarded. The precipitate was suspended in distilled water. 3 M sodium acetate was added, and the mixture vortexed. Following addition of cold absolute ethanol, the mixture was vortexed again, and the suspension again allowed to precipitate overnight at –20°C. After the centrifugation, the precipitate was suspended in 10 mM Tris–Cl (pH 7.4), and DNase I (100 µg mL$^{-1}$; NEB) and RNase (25 µg mL$^{-1}$; NEB) added. The mixture was incubated at 37°C for 4 hr, then placed in a 65°C water bath for 30 min. Ninety percent phenol preheated to 65°C was added, and allow to set at 65°C for 15 min. The mixture was placed in an ice bath to cool, and then centrifuged at 6000×*g* for 15 min. The top aqueous layer was removed, and the phenolic layer re-extracted with an equal volume of distilled water. This sample was again incubated at 65°C for 15 min and then placed in ice water. After centrifugation at 6000×*g* for 15 min, the two aqueous layers were combined and dialyzed against multiple changes of distilled water over 2 days.

To test whether RIF-1 was a peptidoglycan, *A. machipongonensis* peptidoglycan was isolated using a method adapted from *de Jong et al. (1992)*, *A. machipongonensis* cell pellet was washed with 0.8% NaCl. The cells were resuspended in hot 4% SDS, boiled for 30 min, and then incubated at room temperate overnight. The sample was boiled for an additional 10 min and then centrifuged at 15,000×*g* for 15 min at room temperature. The pellet was washed four times with water and resuspended in water. The sample was digested for mutanolysin (10 µg mL⁻¹; Sigma) overnight at 37°C. The enzyme was inactivated by incubation at 80°C for 20 min.

## Isolation and purification of RIF-1 from *A. machipongonensis*

*A. machipongonensis* was cultured in seawater complete medium (16×1 L) at 30°C for 2 days. The cells were harvested by centrifugation and extracted with $CHCl_3$:MeOH (2:1, 4 L). The organic extract was filtered, dried over sodium sulfate ($Na_2SO_4$), and concentrated to give approximately 4 g crude lipid extract. The crude extract was dissolved in a minimum amount of $CHCl_3$:MeOH (2:1), and purified by preparative high performance liquid chromatography (HPLC). All solvents were purchased from Fisher Scientific unless otherwise noted. Preparative reversed phase HPLC (RP-HPLC) was performed on an Agilent Technologies 1200 Series HPLC using a Phenomenex Luna 5 µm C8(2) 100 Å 250×21.2 mm column. Isolation of RIF-1 continued with a crude fractionation in which compounds were eluted at 10 mL min⁻¹ in a gradient of solvents A (0.1% $NH_4OH$ in water) and B (0.1% $NH_4OH$ in methanol): 65% B increasing to 100% B over 30 min, isocratic at 100% B for 1 min. before returning to 65% B and re-equilibrating over 10 min. Fractions were analyzed by low-resolution mass spectrometry (LC-MS) on an Agilent 6130 LC/MS using a Phenomenex Gemini-NX 5 µm C18 110 Å 100×2 mm column. The next stage involved a higher resolution separation in which compounds were eluted at 0.5 mL min⁻¹ in a gradient of solvents A (0.1% $NH_4OH$ in water) and B (0.1% $NH_4OH$ in methanol): 65% B increasing to 100% B over 30 min, isocratic at 100% B for 1 min before returning to 65% B and re-equilibrating over 3 min and those which contained a mass peak corresponding to RIF-1 ([M-H]=606.4) were combined and concentrated. This material was then purified by preparative TLC (1 mm, silica gel 60), eluted with $CHCl_3$:MeOH:AcOH:$H_2O$ (100:20:12:5, Rf=0.5). RIF-1 was visualized by staining with ammonium molybdate in 10% $H_2SO_4$. The portion of the plate (Fraction F; *Figure 3—figure supplement 1*) that induced colony formation and contained RIF-1 (LC/MS: [M-H] 606) was scraped off after RIF-1 was visualized by staining with ammonium molybdate in 10% $H_2SO_4$, and the silica was extracted with $CHCl_3$:MeOH (5:1). This material was further purified by preparative TLC on a 250 µm TLC plate (silica gel 60), eluted with $CHCl_3$:MeOH:AcOH:$H_2O$ (100:20:12:5). From 16 L of *A. machipongonensis* culture, approximately 50 µg RIF-1 was obtained in sufficient purity. The entire process, from growth of the cells to isolation of pure RIF-1, was repeated nine times in order to obtain approximately 0.7 mg RIF-1 from a total of 160 L of *A. machipongonensis* culture.

High Resolution Mass Spectrometry (HRMS) was carried out by Ted Voss at the WM Keck Foundation Biotechnology Resource Laboratory at Yale University on a Bruker 9.4T FT-ICR MS. RIF-1 was dissolved in 200 µL DMSO-$d_6$ and transferred into a 3 mm NMR tube. ¹H, TOCSY, gCOSY and dqfCOSY were recorded on a Varian Inova 600 spectrometer. HMQC and gHMBC experiments were performed on a Bruker Advance (sgu) 900 MHz and Varian Unity Inova 600 MHz equipped with a cryoprobe, respectively. Chemical shifts are reported in ppm from tetramethylsilane with the solvent resonance resulting from incomplete deuteration as the internal standard (DMSO: δ 2.50). Data are reported in *Table 3* as follows: chemical shift, multiplicity (s = singlet, d = doublet, t = triplet, q = quartet, br = broad, m = multiplet), coupling constants, and integration. Optical rotation was measured on a Jasco P-2000 digital polarimeter with a sodium lamp at 21.4°C. Unless otherwise noted, all solvents and reagents were purchased from VWR or Fisher and used without further purification.

**¹H NMR (600 MHz, DMSO)** and **¹³C NMR (600 MHz, DMSO)**: see *Table 3*. **Optical rotation:** $[\alpha]_D^{21.4}$ +6.4 (*c*=0.07, MeOH). **HRMS** *m/z* calcd for $C_{32}H_{64}NO_7S$ (M-H): 606.44035. Found: 606.44027 (M-H)⁻. **MS/MS analysis:** A major fragment derived from *m/z*=606 (M-H) in the MS/MS spectrum of RIF-1 corresponds to amino-sulfonic acid, *Figure 3—figure supplement 2*. **HRMS/MS** *m/z* calculated for $C_{17}H_{36}NO_5S$ (M-H): 366.23142. Found: 366.2310 (M-H)⁻.

## Quantification of RIF-1 levels in conditioned medium

Conditioned medium was prepared from *A. machipongonensis* grown in seawater complete medium (750 mL) at 30°C for 2 days. The conditioned medium was lyophilized and extracted with $CHCl_3$:MeOH

(2:1; 78 mL). The organic extract was filtered, further extracted with $CHCl_3$ (60 mL×2), and filtrates were combined and concentrated to dryness under vacuum. The crude extract was suspended in 5 mL 50% $MeOH:H_2O$ and was passed through a C-18 SPE (1 g) column. The open column was then washed with 10 mL 90% $MeOH:CHCl_3$. The organic eluate was concentrated and dissolved in 3 mL $CHCl_3:MeOH$ (2:1) for LC/MS analysis. The chromatography was carried out using an Agilent 6130 Quadrupole LC/MS system with a C18 reverse-phase column (4.6×100 mm; Phenomenex Luna; 5 μ) for 30 min in a linear gradient from solvent A (60% methanol/water with 0.1% ammonium hydroxide) to solvent B (100% methanol with 0.1% ammonium hydroxide). The RIF-1 was detected in the conditioned medium at a concentration of 80 ng $L^{-1}$. The purified RIF-1 was used as the standard (*Figure 4—figure supplements 1–3*).

## Activity profile of RIF-1

The potency of pure RIF-1 was determined using the quantitative bioassay for rosette colony development. Briefly, 100 ug of pure RIF-1 was solubilized in 100 μL DMSO and this 1 g $L^{-1}$ stock was stored at -80°C. For each experiment, serial dilutions ranging from $10^{-1}$ g $L^{-1}$ down to $10^{-17}$ g $L^{-1}$ were made in DMSO. 2 μL of each dilution was premixed with 1 mL of fresh cereal grass infused seawater (*King et al. 2003*) to avoid precipitation of RIF-1 and the premixed RIF-1 dilution was then added to 1 mL RCA cultures to yield final concentrations ranging from $10^{-3}$ to $10^{-20}$ g $L^{-1}$, equivalent to 1.6×$10^9$ fM down to 1.6×$10^{-8}$ fM. The percentage of rosette colonial cells was determined as described above in three independent cell lines in triplicate. From the percent rosette colony development, a bell-shaped dose-response model was determined to be the nonlinear regression curve of best fit determined using GraphPad Prism 5 statistical software.

# Acknowledgements

We thank Michael Fischbach, Richard Losick, and Russell Vance for critical reading of the manuscript. NK is a Fellow in the Integrated Microbial Biodiversity Program of the Canadian Institute for Advanced Research.

# Additional information

## Competing interests

JC: Reviewing Editor, *eLife*. The remaining authors have no competing interests to declare.

## Funding

| Funder | Grant reference number | Author |
| --- | --- | --- |
| Gordon and Betty Moore Foundation Marine Microbiology Initiative | | Nicole King |
| National Institutes of Health | F32 GM086054 | Rosanna A. Alegado |
| National Institutes of Health | F32 GM089018 | Laura W. Brown |
| National Institutes of Health | R01 GM086258 | Jon Clardy |
| National Institutes of Health | R01 GM099533 | Jon Clardy, Nicole King |
| National Institutes of Health | T32 HG00047 | Stephen R. Fairclough |

The funders had no role in study design, data collection and interpretation, or the decision to submit the work for publication.

## Author contributions

RA: Conception and design, Acquisition of data, Analysis and interpretation of data, Drafting or revising the article; LB: Conception and design, Acquisition of data, Analysis and interpretation of data, Drafting or revising the article; SC: Acquisition of data, Analysis and interpretation of data, Drafting or revising the article; RD: Acquisition of data, Analysis and interpretation of data; RZ: Acquisition of data, Analysis and interpretation of data; SF: Acquisition of data, Analysis and interpretation of data; JC: Conception and design, Analysis and interpretation of data, Drafting or revising the article; NK: Conception and design, Analysis and interpretation of data, Drafting or revising the article

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
