## [Author Response]

We are grateful to the reviewers and editors for their thorough consideration of our manuscript and constructive recommendations for revision. In response to comments made in the general assessment of the manuscript, our revisions include:

1) *Discussion of other signaling systems with ultrasensitive receptors.* Namely, we draw the attention of readers to the well-described silkworm moth sex pheromone signaling system, in which 4 fM bombykol (the silkworm moth sex pheromone) is sufficient to induce a behavioral response in males. The potency of bombykol is comparable to that of RIF-1, which is active at concentrations ranging from 10-2 fM to 107 fM.

*2) An expanded discussion of the difference in activity between purified RIF-1 vs. the sphingolipid-enriched fraction.* We clarify that the difference in activity could be due either to delivery issues (i.e. a requirement for RIF-1 to be delivered in the context of the bacterial membrane) or to the absence of other currently unidentified *A. machipongonensis* molecules that either amplify RIF-1 signaling or might independently induce colony development in *S. rosetta.* In addition, we state that “We hypothesize that RIF-1 may be released into the environment in membrane vesicles, which have been described in Gram-negative bacteria and Bacteroidetes and that additional membrane constituents might be required for the full potency of RIF-1.”

3) A more detailed discussion of future directions for this research, including (as pointed out by the reviewers) the need to determine the three-dimensional structure of RIF-1. Note that defining the stereochemistry will require synthesizing several different possible stereoisomers of RIF-1, and that there are no published total syntheses of *any* sulfonolipids in the literature.